# Ultrasound-Assisted Extraction of Alginate from *Fucus vesiculosus* Seaweed By-Product Post-Fucoidan Extraction

**DOI:** 10.3390/md22110516

**Published:** 2024-11-14

**Authors:** Viruja Ummat, Ming Zhao, Saravana Periaswamy Sivagnanam, Shanmugapriya Karuppusamy, Henry Lyons, Stephen Fitzpatrick, Shaba Noore, Dilip K. Rai, Laura G. Gómez-Mascaraque, Colm O’Donnell, Anet Režek Jambark, Brijesh Kumar Tiwari

**Affiliations:** 1UCD School of Biosystems and Food Engineering, University College Dublin, Belfield, 4 Dublin, Ireland; ming.zhao@ucd.ie (M.Z.); shanmugapriya.karuppusamy@ucd.ie (S.K.); shaba.noore@ucdconnect.ie (S.N.); colm.odonnell@ucd.ie (C.O.); 2Teagasc Ashtown Food Research Centre, Teagasc, 15 Dublin, Ireland; dilip.rai@teagasc.ie (D.K.R.); brijesh.tiwari@teagasc.ie (B.K.T.); 3BiOrbic Bioeconomy SFI Research Centre, University College Dublin, 4 Dublin, Ireland; saravana.periaswamysivagnanam@mtu.ie; 4Department of Biological Sciences, Munster Technological University, T12 P928 Cork, Ireland; 5Nutramara Ltd., V92 FH0K Tralee, Ireland; h.lyons@nutramara.com (H.L.); s.fitzpatrick@nutramara.com (S.F.); 6Teagasc Food Research Centre, Moorepark, Fermoy, P61 C996 Cork, Ireland; laura.mascaraque@teagasc.ie; 7Food Technology and Biotechnology, University of Zagreb, HR-10000 Zagreb, Croatia; anet.rezek.jambrak@pbf.unizg.hr

**Keywords:** seaweed, byproduct, alginate, ultrasound, RSM

## Abstract

The solid phase byproduct obtained after conventional fucoidan extraction from the brown seaweed *Fucus vesiculosus* can be used as a source containing alginate. This study involves ultrasound-assisted extraction (UAE) of alginate from the byproduct using sodium bicarbonate. Response surface methodology (RSM) was applied to obtain the optimum conditions for alginate extraction. The ultrasound (US) treatments included 20 kHz of frequency, 20–91% of amplitude, and an extraction time of 6–34 min. The studied investigated the crude alginate yield (%), molecular weight, and alginate content (%) of the extracts. The optimum conditions for obtaining alginate with low molecular weight were found to be 69% US amplitude and sonication time of 30 min. The alginate extracts obtained were characterized using Fourier transform infrared (FTIR) spectroscopy, thermogravimetric analysis (TGA), and differential scanning calorimetry (DSC). Ultrasound-assisted extraction involving a short treatment lasting 6–34 min was found to be effective in extracting alginate from the byproduct compared to the conventional extraction of alginate using stirring at 415 rpm and 60 °C for 24 h. The US treatments did not adversely impact the alginate obtained, and the extracted alginates were found to have similar characteristics to the alginate obtained from conventional extraction and commercial sodium alginate.

## 1. Introduction

Seaweeds are a major source of high-value biomolecules, including polysaccharides, proteins, phenolics, and pigments, which are utilized in various food, feed, agriculture, biomedical, pharmaceuticals, nutritional, and cosmetic applications. Seaweed polysaccharides have been widely investigated [1]. The major polysaccharides found in the cell walls of brown seaweeds are alginate and fucose containing polysaccharides such as fucoidan, which make up 18–48% and 2–10% of the dry mass, respectively [2]. Due to the potential of valuable bioactive compounds in seaweeds, there is an immense opportunity for further exploration of seaweed species which have not been adequately investigated to date [3].

With the current global issues, such as food insecurity and resource scarcity for the growing population, it has become necessary for agricultural sectors to be aware of waste management after processing. To address this issue, there is a push towards a circular economy approach that looks at waste as a secondary resource [4]. In the phycocolloids industry, only 15–30% of the total dry mass of seaweed is utilized, with the remaining 70–85% amount being degraded during extraction and treated as waste material [5].

It is not possible to extract 100% of biomolecules during extraction processes, and thus, alternative uses for seaweed residues are being explored. For example, the seaweed residues can be utilized for bioethanol production [6]; for removing chromium from aqueous solutions [7]; and for producing seafood-like flavor [8], agricultural fertilizers [9], and animal feed [10]. The utilization of the residual biomass can be carried out in a sequential process involving the extraction of sap and processing of the residual biomass for lipid extraction, followed by protein and cellulose extraction [11]. Baghel, Trivedi, and Reddy [5] extracted pigments from the seaweed *Gracilaria corticata*. The residual biomass obtained was further treated with chloroform–methanol solvent to extract lipids, followed by extraction of agar and enzymatic hydrolysis to form bioethanol. The final product was used as a soil conditioner.

The use of the chemicals during these processes must also be carefully considered to determine the safe utilization of the byproducts. For example, in seaweed processing, formaldehyde is used as a pretreatment for removal of phenols, chloroform is used for binding colored pigments, and HCl is used to convert the insoluble alginic salts to alginic acid [12].

However, due to their high toxicity, these solvents are not suitable for biomass that is processed for food/feed purposes. Thus, it is important to consider the safety of the chemicals used throughout the extraction processes. This has led to a shift towards novel technologies and green solvents that have better extraction efficiencies and also reduce/eliminate the use of harsh chemicals. Some of these technologies include UAE, microwave-assisted extraction (MAE), enzyme-assisted extraction (EAE), supercritical fluid extraction (SFE), etc. A combination of these technologies can also be used for a more efficient extraction process [13].

Alginates are sodium, magnesium, or calcium salts of alginic acid [2] and consist of (1, 4)-linked β-D-mannuronic acid and α-L-guluronic acid residues [14]. Owing to their properties, including thickening, gel-forming ability and stability, film formation, non-toxicity, biodegradability, and ease of processing, alginates are widely used as additives in the food industry [15]. Brown seaweeds are the major source of alginates [16], while alginates can also be obtained from *Azotobacter* and *Pseudomonas* bacteria [17].

High temperatures and long extraction times are desirable to obtain high yields of alginate. From an industrial point of view and in order to maintain the required Mw, a storage temperature of 25 °C and short extraction times are generally preferred [18]. The conversion of insoluble forms of alginate in the seaweed cell walls to a sodium salt (soluble form) can be completed at the beginning of the extraction process, and subsequently, they can be purified using acidic and alkaline solutions [19]. Mw is one of the most important structural features of polysaccharides. Several reports have suggested that their reported antioxidant potency is mainly associated with the Mw of polysaccharides [20]. Many physical, chemical, and enzymatical methods have been used to degrade polysaccharides and reduce their Mw. However, the applications of alginate have been greatly limited due to their high Mw and low bioavailability. The degradation of polysaccharides from high Mw to low Mw poly- or oligo-saccharides increases the bioavailability and absorption of drugs to enhance the efficacy of polysaccharides [21]. UAE has been demonstrated to enhance the extraction of a wide range of biomolecules from seaweeds. However, only limited studies have investigated the extraction of alginate using US [22,23,24]. In this study, fucoidan, a high-value biomolecule, was extracted from seaweed *Fucus vesiculosus* on the industrial scale. The process involved using dry and milled seaweed along with a green extraction solvent (confidential) under conventional extraction conditions of 80 °C for 2 h [25]. The mixture was then filtered to separate the extract and the residue/byproduct. The byproduct obtained was dried and stored to further study the comparison of UAE and conventional extraction approaches to obtain alginate from a byproduct obtained from the large-scale production of fucoidan using *Fucus vesiculosus* seaweed. The impact of selected US amplitudes and sonication treatment times on crude alginate yield, alginate content (%), and Mw were also investigated.

## 2. Results

The crude alginate samples obtained using different extraction conditions were analyzed to determine crude alginate yield, alginate content, and Mw. The samples were also characterized using TGA, DSC, and FTIR.

### 2.1. Proximate Analysis of Fucus vesiculosus Byproduct

Proximate analysis showed that the byproduct contained 1.59 ± 0.07% protein (using 4.17 factor), 2.21 ± 0.58% fat, 31.10 ± 0.21% ash, 4.41 ± 0.15% moisture, and 60.69% total dietary fiber, which can be calculated by subtracting the sum of other components, i.e., 39.31%, from 100%. The alginate content of the sample was 10.71 ± 0.29%.

### 2.2. Experimental Design and Statistical Analysis

Table 1 presents the experimental conditions investigated along with the predicted and experimental values for crude alginate yield (%), alginate content (%), and Mw (kDa) of the samples obtained from UAE using *Fucus* byproduct. The crude alginate yield ranged from 0.020% (30% US amplitude and 10 min sonication time) to 6.19% (55% US amplitude and 20 min sonication time). The crude alginate samples obtained were analyzed for alginate content, and a maximum of 78.22% was observed for a sample treated with 55% US amplitude for a 6 min sonication time, while a minimum value of 39.76% was observed for a sample treated with 30% US amplitude for 30 min. The maximum value of Mw of 120 kDa was achieved for 55% US amplitude and 34 min sonication, while the lowest value of 95.63 kDa was observed for 30% amplitude and 10 min treatment time. The adequacy and significance of the models were evaluated by analysis of variance (ANOVA) using a Fisher’s F-test (Table 2). The CCD design matrix and the results of the RSM experiment to determine the individual and interactive effects of the independent variables of US amplitude and sonication time are shown in Table 1. The *p* value, F value, and R squared values indicate the significance of the model and the effects of the independent variable on the response variables. A quadratic model was found to have the best fit. An US amplitude of 68.86% and sonication time of 30 min were found to be the optimum extraction conditions for obtaining crude alginate with the maximum alginate content (%). According to Table 2, the quadratic model for alginate content (%) had the best fitness (F-value of 62.69) compared to models developed for crude alginate yield and Mw. The independent variable A (% ultrasonic amplitude) had a significant effect on alginate content (%), but no significant effect on crude alginate yield and Mw. The sonication treatment time had significant effects on all responses, as indicated by *p* values of 0.0224, 0.0004, and 0.0016 for crude alginate yield, alginate content, and Mw, respectively.

Table 3 presents the adjusted R^2^ values of 0.72 for crude alginate yield, 0.96 for alginate content, and 0.63 for Mw. The adjusted R^2^ generally ranges between 0 and 1 and shows the amount of data variations explained by the model. The fitness demonstrated by the model was strong for alginate content (close to 1), while for the other responses, it was slightly lower, indicating adequacy for the dependent variables. The lack of fitness was not significant in all responses, indicating the adequacy of the model.

### 2.3. Model Fitting

Table 4 presents second-order polynomial equations that express the relationship between process variables and responses in terms of coded factors. These models were obtained via multiple regression analysis and provide information on the US amplitude and treatment time needed to achieve the optimum crude alginate yield (%), alginate content, and Mw.

### 2.4. Impact of Ultrasound Amplitude (%) and Sonication Treatment Time on Alginate Yield, Alginate Content and Mw

Figure 1a–c show the effects of US amplitude and sonication time on crude alginate yield (%), alginate content (%), and Mw, respectively. The response surface plots indicate that an increase in US amplitude and sonication time leads to an increase in the quantity of alginate obtained. Crude alginate yield (%) (Figure 1a) was also found to increase at higher US amplitudes and with longer sonication times.

The extraction of bioactive compounds from propolis was investigated [26], and an increase in extraction yield was observed when the US amplitude level was increased. The compression and rarefaction cycles depend on the ultrasonic wave amplitude, i.e., at higher the amplitude, the greater the number of cavity formations, leading to the maximum extraction yield [27].

An increase in US power from 75 to 150 W increased the extraction efficiency of alginate from *Sargassum binderi* and *Turbinaria ornata* [22]. They also reported that the extraction yield was influenced by pH, algae/water ratio, and exposure time to US at 25 kHz. The maximum extraction yield was observed for sonication treatment of 40 min at pH 12 using a 10 g/L algae (dw)/water ratio. UAE was demonstrated to double the alginate yield over 30 min compared to both the conventional extraction method (90 °C, 2 h stirring) and a conventional method assisted with US for 15–30 min.

With a high US amplitude, the crude alginate yield (%) increased, which may be attributed to the impact of cavitation bubbles formed due to US. The collapse of the cavitation bubbles creates strong turbulence, collisions between particles, and disturbances in the pores of the biomass. The cavitation near the solid–liquid interfaces also produces liquid jets that degrade the seaweed matrix and penetrate the surface cavities, thereby enhancing mass transfer [28].

US treatment (35 kHz, 30 min, and 50% ethanol) was reported [29] to enhance the extraction of a range of biomolecules, including phenolics, phlorotannins, fucoidan, and other polysaccharides from brown seaweeds. In the current study, the opposite effect was observed for alginate content (Figure 1b), namely that a lower alginate content was observed with an increase in sonication time. Meanwhile, an increase in ultrasonic amplitude and sonication time led to the extraction of higher-Mw alginate (Figure 1c). Ultrasound may cause degradation and rearrangement, which may lead to extraction of high-Mw alginate samples.

The impact of ultrasonic frequencies on the properties of sodium alginate was investigated [30]. Sodium alginate solutions were treated with US at 0.25 W/cm^2^ and 50 °C for 20 min using four different frequencies (28, 40, 50, and 135 kHz). The Mw of was found to change with the ultrasonic frequency, showing both degradation and rearrangement effects. The Mw decreased at low frequencies (28 and 40 kHz), but increased at high frequencies (50 and 135 kHz). This was due to the different impacts of US on the structure and composition of sodium alginate. The authors also suggested that US enhanced the formation of covalent bonds or non-covalent interactions such as electrostatic forces among sodium alginate molecules.

### 2.5. Comparison of Optimized UAE with Conventional Extraction Method for Alginate Yield, Alginate Content, Mw, and Color

In the current study, the optimized UAE conditions of 69% US amplitude for 30 min were used in the extraction processes of O1 and O2 samples (samples obtained by using optimized extraction conditions) and resulted in 6.71 and 6.94% crude alginate yields, respectively (Table 5). The Mw and alginate content (%) of the samples obtained were 105.24 and 105.80 kDa and 62.48 and 63.17%, respectively. These results are comparable to the values obtained for the conventionally extracted samples CA and CB.

Color analysis showed that samples obtained using US treatment were slightly more yellowish compared to commercial sodium alginate, but lighter than conventionally extracted samples. The L, a, and b values for the commercial sample were 94.95, −0.86, and 11.73, respectively. The ΔE values of the optimized alginate samples were 26.14 and 24.31 for O1 and O2, respectively. Prolonged exposure to moderate or high heat can lead to darker-colored alginate samples.

Since conventional extraction involves using a high temperature for 24 h, all the conventionally treated samples in this study were dark brown. These results are in accordance with the study by [31], which investigated the use of conventional extraction of pectin from pomelo peel using acid (88 °C, 141.4 min), subcritical water (120 °C, 20–100 min), and a US–microwave combination (40 kHz US, 643 W microwave, 27.5 min). The authors reported that shorter treatment times led to lighter-colored samples.

Overall, UAE is considered a “greener” method because it reduces the solvent and time requirements, thereby conserving energy [22]. Therefore, optimized UAE (69% US, 30 min) has potential to replace the conventional extraction method (60 °C stirring for 24 h).

### 2.6. Characterization Studies of Standard Sodium Alginate, Sodium Bicarbonate, and Alginate Obtained Using Optimized Ultrasound-Assisted Extraction Conditions

#### 2.6.1. Thermal Properties (TGA and DSC) of Alginate Samples

TGA is used to determine changes in mass during thermal treatment, with weight changes indicating material degradation. DSC measures variations in a material’s heat capacity due to changes in temperature, detecting changes in the polymer structure, such as melting or phase transition [32]. The TGA and DSC curves of reference sodium bicarbonate and commercial alginate, as well as samples extracted using optimized US conditions, were compared (Figure 2) over a temperature range of 0–340 °C.

TGA analysis revealed three stages of mass loss, namely moisture loss, degradation of the alginate structure, and complete thermal degradation. Initially, all the samples exhibited similar decomposition patterns, mainly due to moisture loss. Mass loss was observed to occur in all samples above 100 °C. Between 220 and 260 °C, a mass loss was observed, which was associated with the destruction of glycosidic bonds and corresponded to alginate fragmentation due to polymeric chain breakage. The mass loss curves became linear around 400 °C, which was attributed to the conversion of monomers and fragments of alginate to Na_2_CO_3_. Similar patterns were observed for commercial sodium alginate and alginate samples extracted using optimized US extraction conditions.

These results are in accordance with TGA analysis of sodium alginate [33]. Similar TGA patterns were found when investigating the extraction of alginate from seaweed *Saccharina japonica* using a deep eutectic solvent in subcritical water hydrolysis (150 °C, 19.85 bar, 70% water content, and 36.81 mL/g solid/liquid ratio) [34].

In DSC, changes in the heat flow can be observed during a thermal process, which includes the generation of heat (exothermic) or consumption of heat (endothermic) by the sample. Mass loss occurred in all samples up to 120 °C, endothermic peaks were observed at 160–170 °C in the sodium bicarbonate, and a slight decline was observed in optimized alginate samples, indicating their hydrophilic nature. The sodium bicarbonate present in the optimized alginate samples resulted from the extraction process and lead to similar patterns in both the optimized alginate samples and reference sodium bicarbonate. In contrast, no such peak or similarity was observed in the commercial sodium alginate. In commercial sodium alginate, a broad exothermic peak was observed at 240 °C, potentially due to biopolymer and carbonate decomposition [35]. Similar findings were reported by Saravana, Cho, Woo, and Chun [34], where the authors also reported that weakly bound water was released at 80 °C, resulting in an endothermic peak, followed by an exothermic peak at 225 °C due to polymer breakdown. The thermal transitions were noted and were reported to be mainly due to the loss of water in the alginate, with the weakly bound water lost around 40–60 °C and the hydroxyl-bound water released around 90–120 °C. Overall, both the TGA and DSC curves showed similarity between commercial sodium alginate and alginate obtained with optimized US extraction conditions, indicating that US had no negative impacts on the thermal properties of the alginate extracted.

#### 2.6.2. Fourier Transform Infrared Spectroscopy

The mean FTIR reflectance spectra of crude alginate extracted using UAE and conventional extraction treatments (i.e., TCA and TCB) and a commercial sodium alginate sample were measured over the FTIR wavenumber range of 450–1800 cm^−1^ (Figure 3). It can be observed that the spectral features of the extracted samples are very similar to that of a commercial sodium alginate sample. Therefore, sodium alginate is likely to be the main polysaccharide contained in the extracted alginate samples from the current study. There are four specific spectral features related to the functional groups of sodium alginate, which can be observed at 809, 1023, 1404, and 1591 cm^−1^ [36].

The bands in Figure 3 at both 809 and 1023 cm^−1^ are related to skeletal –C–C~ and –CO stretching of pyanose rings, especially the band around 1025 cm^−1^ assigned to C–O–C of glycosidic linkage of polysaccharides [37]. The band at 1404 cm^−1^ is associated with C–OH deformation vibration with the contribution of O–C–O symmetric stretching vibration of the carboxylate group [38], while the band at 1591 cm^−1^ is strongly related to hydrogen-bonded COO− [39] or to carboxylate O-C-O asymmetric stretching vibration and C=O asymmetric stretching vibration of uronic acids [18]. In addition, the bands around 1025–1030 cm^−1^ are associated with the C-O group [40,41], and the bands in the anomeric region of 950–750 cm^−1^ are associated with alginate characterization [41].

## 3. Materials and Methods

### 3.1. Biological Material

Large-scale extraction of fucoidan using 750 kg of *Fucus vesiculosus* seaweed was carried out by Nutramara Ltd. (Tralee, Ireland) using the extraction method. Some of the seaweed byproduct was collected, and the samples were freeze dried at 0.5 mBar for 3 days using a freeze dryer (Lyovapor^TM^, L-300, Buchi, Flawil, Switzerland), then milled to a 1 mm particle size. Samples were packed and stored at 4 °C until further use.

### 3.2. Chemicals

Sodium bicarbonate (BioSciences, Torrace, CA, USA), sodium chloride (NaCl) (code 131651, PanReac Applichem ITW reagents, Castellar del Vallès, Spain), sodium alginate (Sigma-Aldrich, St. Louis, MO, USA), isopropyl alcohol (H625, Romil pure chemistry, Cambridge, UK), Na_2_SO_4_ (Alfa Aesar, Haverhill, MA, USA), EDTA (Sigma-Aldrich, St. Louis, MO, USA), and NaOH (Acros Orgnaics, The Hague, The Netherlands) were procured. Distilled water from Millipore Milli-Q Purification System (Southern Scientific Instruments, Granite Quarry, NC, USA) was used for all the extraction experiments.

### 3.3. Proximate Analysis of Byproduct

Proximate analysis included the determination of fat, ash, protein, and moisture content. The fat content was determined according to the AOAC Official Method (2008) using an Oracle fat analyzer (Oracle, CEM Corporation, Matthews, NC, USA) based on NMR technology. Ash content was determined according to ISO2171 using a Pyro Milestone Microwave muffle furnace (Milestone Srl, Milan, Italy). Moisture was determined according to ISO2171 using an M-Series Sanyo Gallenkamp air oven (Leicester, UK) kept at 105 °C overnight. Protein content was determined according to AOAC Method 968.06 (2005) using a nitrogen analyzer (FP328 Leco Instrument; Leco Corporation, St. Joseph, MI, USA) based on the Dumas principle (N × 6.25).

### 3.4. Ultrasound-Assisted Extraction and Conventional Extraction Method to Obtain Alginate from Byproduct

Dry *Fucus vesiculosus* byproduct (40 g) was mixed with 800 mL of 0.1 M sodium bicarbonate solution and sonicated using an ultrasonic processor (UIP500hdT, Hielscher Ultrasonics GmbH, Teltow, Germany) at 20 kHz and a UIP500hd transducer (Hielscher US Technology^TM^, Teltow, Germany) and probe (Sonotrode BS4d18 (titanium, tip diameter 18 mm, length 125 mm), Germany). US amplitudes were set at 20, 30, 55, 80, and 90% and combined with sonication treatment times of 6, 10, 20, 30, and 34 min (based on experimental design). The temperature was controlled using a circulation water bath at 30 °C.

The treated samples were then centrifuged (6000× *g*, 15 min) using a centrifuge (Sorvall LYNX 6000 Super speed, Thermo Scientific™, Waltham, MA, USA) (Figure 4). Meanwhile, NaCl (2% *w*/*v*) and 1:3 isopropyl alcohol (*v*/*v*) solutions were added to the supernatant and stored overnight at 4 °C, followed by filtration using a muslin cloth. The pellet (crude alginate) was dried at 35 °C for 1 day using an Oven (Plus II 2 Lab OPL225.DT1.C 300C, Gallenkamp, UK).

The term “crude alginate” was used to refer to the pellet, as besides alginate, it may have other biomolecules such as phenols, pigments, etc., in it. The crude alginate yield was calculated using the following Equation (1):(1)$$Crude\;alginate\;yield(\%)=\frac{Weight\;of\;dry\;extract\times 100}{Weight\;of\;dry\;sample}$$

The supernatant was subjected to rotary evaporation (rotavapor R-220, Buchi, Flawil, Switzerland) to recover isopropyl alcohol. The conventional extraction method involved stirring at 415 rpm using an overhead stirrer (VWR VOS 40 digital, Dublin, Ireland) in a water bath (Clifton range NE1-2.5 thermostatic bath, UK) at 60 °C for 24 h.

### 3.5. Experimental Design

The experiment was designed and optimized by employing a central composite design (CCD) in response surface methodology (RSM) using design expert (version 7.1.3) software (stat-ease, Minneapolis, MN, USA). Central composite design, a common multivariate optimization method, was used for optimization of the US amplitude and sonication time to evaluate their effects on measured responses (i.e., crude alginate yield, alginate content and Mw). For the levels of independent variables (i.e., US amplitude and sonication time) selected for this study, the amplitude varied between 20% and 90%, and the sonication treatment time varied from 6 to 34 min. The details of the independent variables and levels selected are shown in Table 6.

### 3.6. Alginate Content Determination

The alginate content (%) of crude alginate samples was determined using a high-performance liquid chromatography (HPLC) system (Agilent 1200 LC system, Agilent Technologies, Santa Clara, CA, USA), along with a guard column (OHpak SB-G 6B, 8 × 50 mm) and a Shodex OHpak SB-804 HQ size exclusion chromatography column (Shodex, Japan). A solution of 0.05 M Na_2_SO_4_ and 0.01 M EDTA was prepared and stirred during magnetic heating followed by pH adjustment to 7 by the addition of NaOH. A flow rate of 0.6 mL/min, temperature of 60 °C, injection volume of 10 μL, and run time of 20 min were selected for HPLC analysis [42]. The results obtained were expressed as % *w*/*w* alginate content.

### 3.7. Alginate Molecular Weight Determination

A high-performance liquid chromatograph coupled with a refractive index (HPSEC -RI) detector (Agilent 1200 LC system, Agilent Technologies, Santa Clara, CA, USA) was used to determine the Mw distribution. Shodex OHpak SB-804 HQ with 6% cross-linked HPSEC carbohydrate column with 8 × 300 mm (length × I.D.) connected in series to a guard column (OHpak SB-G 6B, 8 × 50 mm) (Shodex, Japan) using 0.1% NaCl at a constant flow rate of 0.5 mL/min for 40 min and a 40 °C column temperature was used [43]. The samples (2 mg/mL) were prepared with the 0.1% NaCl and filtered through 0.45 µm PTFE filters (Econo Filter, Agilent Technologies). First, 20 µL of samples were injected into the column using an auto sampler. The determination of Mw was performed by comparison of the retention times with those of pullulan standard from Sigma (Set Mw ~350–700,000, Sigma-Aldrich, St. Louis, MO, USA). A standard curve was developed using different Mw of pullulan. The integration of the peaks was performed using the software Agilent Chemstation (HPSEC 2D Chemstation software). All analyses were performed in duplicate.

### 3.8. Color Analysis of Alginate Samples

The colors of the alginate samples obtained from optimized US extraction conditions and conventional method were determined with a Hunter lab colorimeter (Hunter Lab Ultra Scan Pro, Hunter Associates Laboratory Inc., Reston, VA USA) using Equation (2):(2)$$\Delta E=\sqrt{{\left(L-L_{0}\right)}^{2}+{\left(a-a_{o}\right)}^{2}+{\left(b-b_{o}\right)}^{2}}$$
where “o” is measured as a control (Salgado et al. 2011). Total color difference (ΔE) was calculated from the “L”, “a”, and “b” values, where L represents lightness (whiteness or darkness): +a (red); –a (green) and +b (yellow); −b (blue). L_o_, a_o_, and b_o_ represent readings of the control [44].

### 3.9. Alginate Characterization

Characterization of alginate extracts was carried out using thermogravimetric analysis (TGA), differential scanning calorimetry (DSC), and Fourier-transform infrared spectroscopy (FTIR). The quality characteristics of crude alginate obtained through optimized UAE were compared with commercial sodium alginate thermogravimetric analysis (TGA) and differential scanning calorimetry (DSC). The Fourier-transform infrared spectroscopy (FTIR) method was used for comparing commercial sodium alginate with crude alginate obtained using UAE and the conventional extraction method.

#### 3.9.1. Thermogravimetric Analysis (TGA) and Differential Scanning Calorimetry (DSC)

Thermogravimetric analysis (TGA) is a method used to determine kinetic parameters and to quantify the devolatization rates and curves at different heating rates and schedules [45]. It is commonly used for investigating the thermal stability of membranes [46]. Differential scanning calorimetry (DSC) is used for determining the temperature and thermal effects of phase changes during heating and cooling [47].

TGA and DSC were performed as outlined by [48] with minor modifications on crude alginate obtained for optimized US extraction conditions using a thermogravimetric analyzer (TGA Q500, TGA/DSC 3+, USA). The results were compared with those obtained for commercially available sodium alginate and sodium bicarbonate. The nitrogen flow rate was maintained at 30 mL/min, and the temperature was increased from 25 to 600 °C at a constant heating rate of 10 °C/min.

#### 3.9.2. Fourier-Transform Infrared Spectroscopy (FTIR)

For FTIR measurements, each crude alginate sample obtained from all US treatments and conventional extraction was milled into a powder, and 0.4 g of each sample was compressed into a pellet with a diameter of 1.4 cm. For comparison, a commercial sodium alginate product (0.4 g) (Sigma-Aldrich, W201502, Saint Louis, MA, USA) was also compressed into a pellet. Each sample pellet was measured using a Fourier transform mid-infrared spectrophotometer (Nicolet™ iS5, Thermo Scientific, Madison, WI, USA) with a diamond crystal attenuated total reflectance (ATR) accessory (iD7 ATR, Thermo Scientific, Madison, USA). The measurement was carried out in single-beam reflectance mode, and air blank background calibration was carried out before each measurement. A total of 64 scans were performed on each measurement to acquire the averaged spectrum. The spectral data were converted into absorbance spectra in the wavenumber range of 600–1800 cm^−1^ with a resolution of 2 cm^−1^. Data acquisition was managed using OMNIC software v. 9.2.98 (Thermo Fisher Scientific Inc., Waltham, MA, USA). Each sample pellet was measured six times at three different surface areas on both sides of the pellet. For further spectral analysis, the collected data were exported and imported into MATLAB 2020b (The Mathworks, Natick, MA, USA) for baseline correction and spectral plotting.

### 3.10. Response Surface Modeling

Response surface modeling was carried out to investigate the impact of the extraction variables, namely US amplitude and sonication treatment time, on the responses, namely crude alginate, alginate content, and alginate Mw. Three-dimensional surface plots were used and analyzed for optimization and interaction of the factors. The statistical significance was determined by analysis of variance (ANOVA). Lack of fit was evaluated to assess the model’s significance. The UAE treatments were compared with the conventional extraction samples.

## 4. Conclusions

US can be employed as an efficient method to extract alginate from seaweed byproduct obtained after extraction of fucoidan. All the samples obtained from US treatments for short time treatments times from 6 to 34 min gave comparable results for alginate yield, alginate content, and Mw to alginate samples which were conventionally extracted using 60 °C and 24 h of stirring. The quality parameters and characterization of the samples revealed that US did not negatively impact the properties of the alginate samples. The color of the samples obtained from optimized UAE treatments was similar to a commercial sodium alginate sample. The conventionally extracted samples were darker in color compared to both a commercial sodium alginate sample and optimized UAE treated samples. The use of response surface methodology assisted the determination of optimum UAE parameters and reduced the number of experiments required to investigate the effect of US amplitude and treatment time on alginate extraction. The results obtained in this study demonstrate that a byproduct of fucoidan extraction has commercial potential as a source of high-quality alginate, which also improves the sustainability of the seaweed industry.

## Figures and Tables

**Figure 1 marinedrugs-22-00516-f001:**
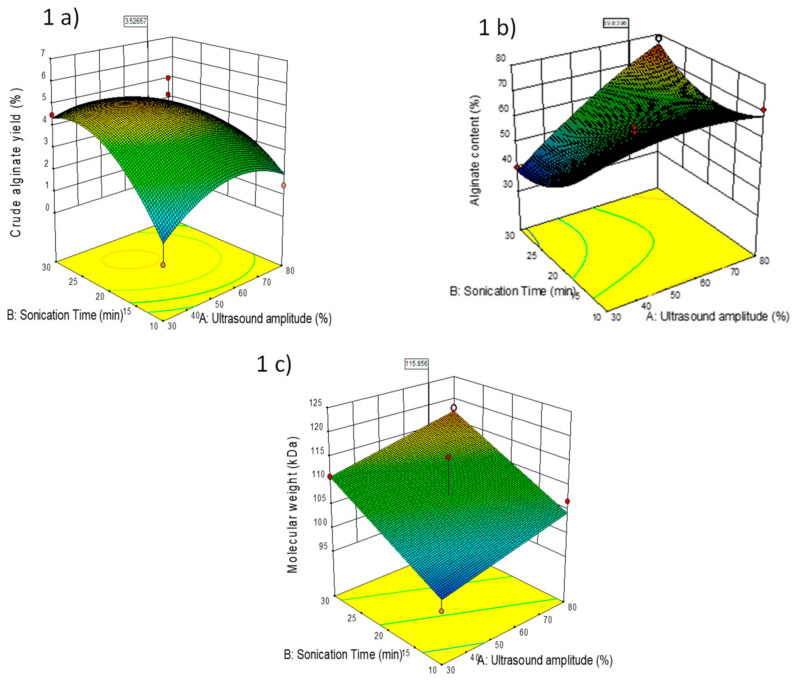
Response surface plots of experimental design showing the effect of ultrasonic amplitude and sonication treatment time on (**a**) crude alginate yield; (**b**) alginate content; and (**c**) Mw.

**Figure 2 marinedrugs-22-00516-f002:**
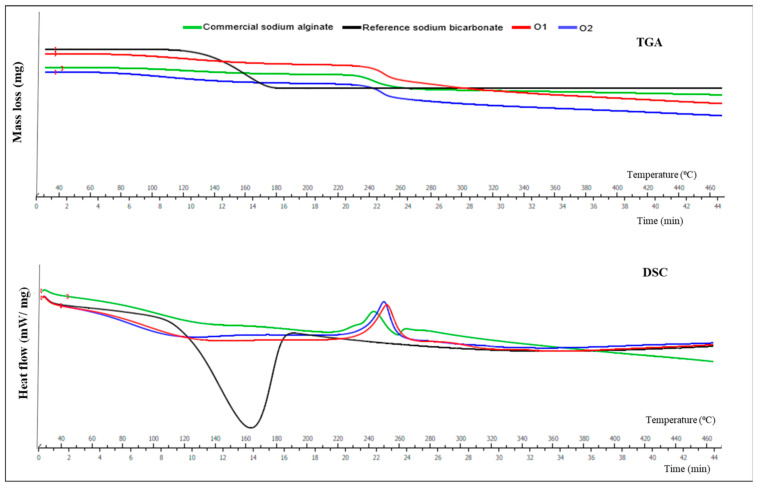
TGA and DSC curves of crude alginate (O1 and O2 samples) obtained with optimum UAE conditions compared with reference sodium bicarbonate and commercial sodium alginate samples.

**Figure 3 marinedrugs-22-00516-f003:**
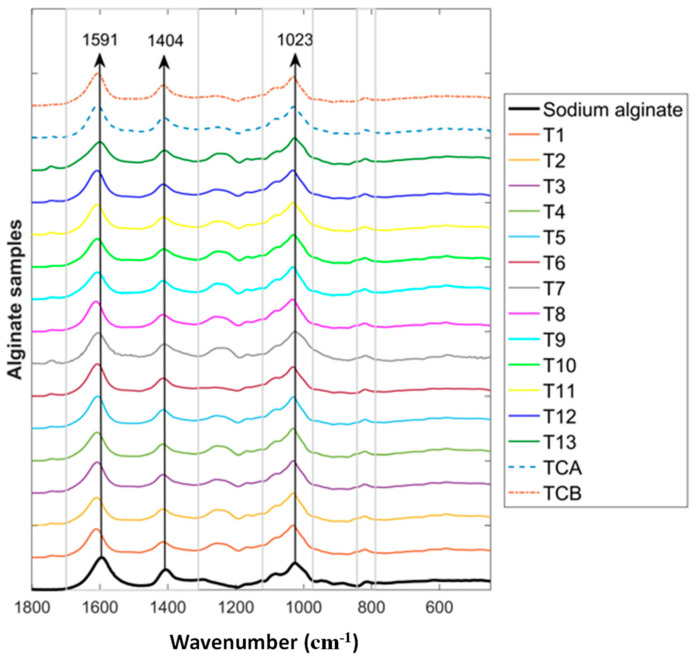
The mean FTIR spectra of alginate samples extracted with UAE (T1–T13), conventional extraction (i.e., TCA and TCB), and a commercial sodium alginate sample.

**Figure 4 marinedrugs-22-00516-f004:**
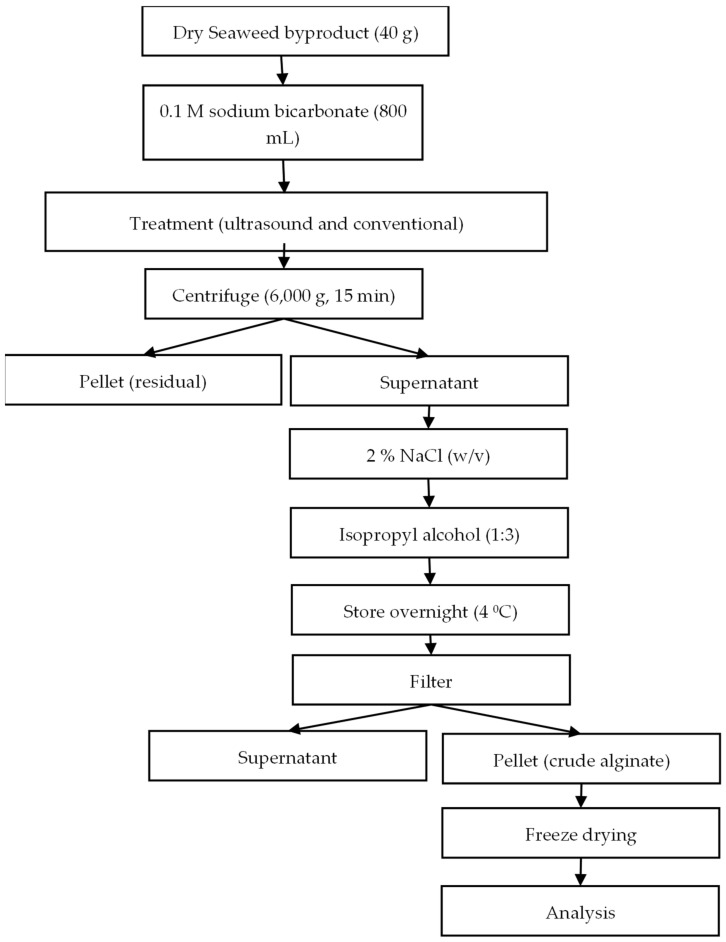
Schematic of alginate extraction workflow using the seaweed byproduct obtained from *Fucus vesiculosus* after fucoidan extraction.

**Table 1 marinedrugs-22-00516-t001:** Experimental design and responses for US-assisted extraction of alginate from *Fucus vesiculosus* byproduct.

			Crude Alginate Yield (% *w*/*w*)		Alginate Content (% *w*/*w*)		Molecular Weight (kDa)	
Treatment (T)	US Amplitude (%)	Sonication Time (min)	A	*p*	A	P	A	*p*
1	90	20	2.16	2.24	63.90	66.15	109.89	112.12
2	30	30	4.49	4.36	39.76	38.31	110.94	110.88
3	80	10	1.42	1.93	70.77	68.37	106.74	104.31
4	20	20	3.45	2.99	37.34	38.94	104.57	103.07
5	55	20	4.17	4.89	57.04	57.80	101.50	107.59
6	80	30	2.79	2.28	78.18	76.28	117.92	117.28
7	55	34	2.93	3.46	63.64	65.21	120.41	116.77
8	55	20	4.30	4.89	58.15	57.80	102.23	107.59
9	55	20	4.31	4.89	59.93	57.80	105.82	107.59
10	55	20	5.46	4.89	55.68	57.80	105.18	107.59
11	55	6	1.70	0.79	78.22	80.50	102.45	98.42
12	30	10	0.020	0.91	69.79	67.84	95.63	97.91
13	55	20	6.19	4.89	58.18	57.80	115.45	107.59

The results are expressed as average (*n* = 2). A: actual values, *p*: predicted values.

**Table 2 marinedrugs-22-00516-t002:** Analysis of variance (ANOVA) for response surface model for crude alginate yield, alginate content, and Mw.

	Crude Alginate Yield (%)		Alginate Content (%)		Mw (kDa)	
Source	F Value	*p*-Value Prob > F	F Value	*p*-Value Prob > F	F Value	*p*-Value Prob > F
Model	7.09	0.0115	62.69	<0.0001	11.32	0.0027
A-US amplitude	0.67	0.4402	128.15	<0.0001	4.44	0.0614
B-Sonication time	8.52	0.0224	40.45	0.0004	18.21	0.0016
AB	2.85	0.1352	60.65	0.0001		
A^2^	10.64	0.0138	8.30	0.0236		
B^2^	15.73	0.0054	68.26	<0.0001		
Lack of Fit	1.11	0.4429	4.13	0.1022	0.32	0.8947

**Table 3 marinedrugs-22-00516-t003:** Lack-of-fit (LOF) test for experimental responses i.e., crude alginate yield, alginate content, and Mw.

	Crude Alginate Yield (%)	Alginate Content (%)	Mw (kDa)
Std. Dev.	0.92	2.40	4.30
Mean	3.34	60.81	107.59
C.V.%	27.50	3.95	4.00
PRESS	24.09	232.80	266.80
−2 Log Likelihood	26.62	51.65	71.40
R-Squared	0.84	0.98	0.69
Adj R-Squared	0.72	0.96	0.63
Pred R-Squared	0.33	0.87	0.56
Adeq Precision	6.58	25.83	9.38
BIC	42.01	67.04	79.10
AICc	52.62	77.65	80.07

**Table 4 marinedrugs-22-00516-t004:** Response surface models of actual and coded factors.

Parameters	Models (Actual Factors)	Models (Coded Factors)
Crude alginate yield (%)	−10.852 + 0.251A + 0.817B − 0.003AB − 0.002A^2^ − 0.014B^2^	4.89 − 0.27A + 0.95B − 0.77AB − 1.14A^2^ − 1.38B^2^
Alginate content (%)	106.036 + 0.098A − 5.611B + 0.037AB − 0.004A^2^ + 0.075B^2^	57.80 + 9.62A − 5.40B + 9.36AB − 2.63A^2^ + 7.53B^2^
Molecular weight (kDa)	87.579 + 0.128A + 0.649B	107.59 + 3.20A + 6.49B

A and B represent US amplitude and sonication treatment time, respectively.

**Table 5 marinedrugs-22-00516-t005:** Effect of optimized US-assisted extraction compared with conventional extraction for alginate yield, alginate content, Mw, and total color difference (ΔE).

Extraction Treatment	Crude Alginate Yield (% *w*/*w*)	Alginate Content (% *w*/*w*)	Molecular Weight (kDa)	Total Color Difference (ΔE)
Optimized UAE treatment (69% US, 30 min) *O1	6.71	62.48	105.24	26.14
Optimized UAE treatment (69% US, 30 min) *O2	6.94	63.17	105.8	24.31
Conventional treatment (60 °C stirring for 24 h) *CA	8.23	66.67	113.8	48.65
Conventional treatment (60 °C stirring for 24 h) *CB	7.81	68.22	106.61	49.33

The results for alginate content, Mw and total color difference are expressed as mean (*n* = 2).

**Table 6 marinedrugs-22-00516-t006:** Independent variables and the coded levels used for optimization of US extraction parameters of US amplitude and sonication treatment time to obtain alginate from *Fucus vesiculosus* byproduct.

Independent Variable	Units	Symbol	Factor Levels				
			−1.412	−1	0	+1	+1.412
US amplitude	%	A	20	30	55	80	90
Sonication treatment time	min	B	6	10	20	30	34

## Data Availability

All data generated or analyzed during this study are included in this published article, further inquiries can be directed to the corresponding author.

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
