# Peer review of "Ultrasound-Assisted Extraction of Alginate from Fucus vesiculosus Seaweed By-Product Post-Fucoidan Extraction"

_marinedrugs, 2024, doi:10.3390/md22110516_

Round 1

Reviewer 1 Report

Comments and Suggestions for Authors

Ummat et al. provided a feasible solution for the utilization of the by-products after extraction of fucoidan from Fucus vesiculosus. The experimental design was reasonable and the results were satisfactory. However, there are still some issues that need to be resolved before publication.

1. The title of the article should be clear and contain the keywords “Fucus vesiculosus” and “fucoidan extraction by-products”.

2. The process of industrial extraction of fucoidan from Fucus vesiculosus and the utilization of the by-products needs to be described in the introduction.

3. Line 297, Delete the first paragraph on materials and methods.

4. Line 334, Why isopropyl alcohol is used to precipitate alginate, not ethanol?

5. Line 348, Why only two factors, amplitude and time, were chosen for response surface experiments?

6. Some experimental methods can be briefly described, such as FT-IR analysis.

7. Line 425, I think this part should follow 3.5 Experimental design.

8. There was little discussion of the results.

9. Incorrect formatting of reference citations in the main text.

Author Response

Comment 1:  The title of the article should be clear and contain the keywords “Fucus vesiculosus” and “fucoidan extraction by-products”.

Response 1: Thank you for pointing this out. We agree with this comment. Therefore, we have changed the title. “Ultrasound-Assisted Extraction of Alginate from Fucus vesiculosus Seaweed By-product Post-Fucoidan Extraction”  

Comments 2: The process of industrial extraction of fucoidan from Fucus vesiculosus and the utilization of the by-products needs to be described in the introduction.

Response 2:  Thank you for the suggestion. We have added the following text in lines 101- 108 in the manuscript: In this study, fucoidan, a high value biomolecule was extracted from seaweed Fucus vesiculosus on industrial scale. The process involved using dry and milled seaweed, along with a green extraction solvent at conventional extraction conditions of 80°C for 2 hours [25]. The mixture was then filtered to separate the extract and the residue/ byproduct. The byproduct obtained was dried and stored to further study comparison of UAE and conventional extraction approaches to obtain alginate from a byproduct obtained from large scale production of fucoidan using seaweed Fucus vesiculosus.

25. Ummat, V.; Sivagnanam, S.P.; Rai, D.K.; O’Donnell, C.; Conway, G.E.; Heffernan, S.M.; Fitzpatrick, S.; Lyons, H.; Curtin, J.; Tiwari, B.K. Conventional Extraction of Fucoidan from Irish Brown Seaweed Fucus Vesiculosus Followed by Ultrasound-Assisted Depolymerization. Sci Rep 2024, 14, 6214, doi:10.1038/s41598-024-55225-z.

Comment 3:  Line 297, Delete the first paragraph on materials and methods.

Response 3: Deleted the first paragraph. Since we think its important to mention these lines, we have moved them to section 3.9. The quality characteristics of crude alginate obtained through optimized UAE were compared with commercial sodium alginate thermogravimetric analysis (TGA), differential scanning calorimetry (DSC). Fourier-transform infrared spectroscopy (FTIR) method was used for comparing commercial sodium alginate with crude alginate obtained using UAE and conventional extraction method.

Comment 4:

Line 334, Why isopropyl alcohol is used to precipitate alginate, not ethanol?

Response 4: Thank you for your question regarding the use of isopropyl alcohol for alginate precipitation instead of ethanol.

We chose isopropyl alcohol based on several key factors:

  1. Literature Support: Our decision is supported by research publications, such as (Bojorges et al., 2022) and (Lee, 1996) etc. which have successfully used isopropanol as the extraction solvent for alginate. These studies highlight the effectiveness of isopropanol in achieving high yields and purity of alginate.
  2. Industrial Practice: Our industrial partners currently use isopropyl alcohol in their extraction processes. To ensure compatibility and optimize their existing procedures, we adopted isopropanol for our experiments. This alignment with industry practices helps in translating our research findings into practical applications.

By using isopropyl alcohol, we aimed to leverage these advantages to enhance the efficiency and effectiveness of our alginate extraction process.

Bojorges, H., Fabra, M.J., López-Rubio, A. & Martínez-Abad, A. (2022). Alginate industrial waste streams as a promising source of value-added compounds valorization. Science of The Total Environment, 838, 156394.

Lee, J. (1996). Role of acetylation on metal induced precipitation of alginates. Carbohydrate Polymers, 29, 337–345.

Comment 5: Line 348, Why only two factors, amplitude and time, were chosen for response surface experiments?

Response 5:  The selection of only ultrasound amplitude and time as the primary factors for our response surface experiments was based on the following considerations:

  1. Primary Influence on Extraction Efficiency: Both ultrasound amplitude and time are known to significantly impact the efficiency of alginate extraction. Amplitude affects the intensity of the cavitation process, while time determines the duration of exposure, both of which are critical for maximizing yield and quality.
  2. Preliminary Studies and Literature Support: Preliminary experiments and literature reviews indicated that these two factors had the most substantial effect on the extraction process. Other variables, such as temperature and solvent concentration, were found to have secondary effects or were kept constant to isolate the impact of amplitude and time.
  3. Experimental Design and Feasibility: Focusing on two primary factors allowed for a more manageable and precise experimental design. This approach ensured a thorough investigation of the interaction between amplitude and time, providing clear insights into their combined effects on alginate extraction.
  4. Optimization Goals: The primary goal was to optimize the ultrasound-assisted extraction process by fine-tuning the most influential parameters. By concentrating on amplitude and time, we aimed to develop a robust and efficient extraction protocol that could be easily scaled up for industrial applications.

Comment 6:  Some experimental methods can be briefly described, such as FT-IR analysis.

Response 6: Thank you for your feedback. While some experimental methods can be briefly described, we have provided a detailed process for the FT-IR analysis. This level of detail is necessary to ensure the method can be accurately replicated by other researchers. Therefore, we believe the current description is appropriate.

Comment 7:  Line 425, I think this part should follow 3.5 Experimental design.

Response 7: Thank you for your suggestion. The details regarding the measurement of each sample pellet and the subsequent data analysis in MATLAB are specific to the FT-IR analysis. Therefore, it is most appropriate to include this information within the FT-IR section to maintain clarity and context. Line 425: Each sample pellet was measured six times at three different surface areas on both sides of the pellet.  For further spectral analysis, the collected data was exported and imported in MATLAB 2020b (The Mathworks, Natick, MA, USA) for baseline correction and spectral plotting.

Comment 8:  There was little discussion of the results.

Response 8: Thank you for your feedback. We acknowledge the importance of a comprehensive discussion of results. However, it is important to note that there is limited literature available on ultrasound-assisted extraction of alginate from seaweed/ seaweed by-product. Most existing studies focus on the extraction of alginate from seaweed, primarily exploring the effects of pH and solvent. Given these constraints, we have made every effort to present as thorough a discussion as possible based on our findings. We believe that our current discussion adequately covers the significant aspects of our research within the context of the available literature. We appreciate your understanding and consideration.

Comment 9: Incorrect formatting of reference citations in the main text.

Response 9: Thank you for pointing it out. The corrections have been made throughout the manuscript. For example in lines 186, 187, 191, 206, 214, 276 and 280. The corrections have been marked red.

Reviewer 2 Report

Comments and Suggestions for Authors

The manuscript reported the ultrasound assisted extraction of alginate from seaweed biomass residues and it is intresting for readers. Therefore, some major revisions needed to be addressed.

1. Figure 1 presented unsolid results of response surface plots and this section needed to be repeated.

2. The recovery ratio, yield and purity of alginate should be described.

3. The alginate should be characterized structurally to confirm its composition, sugar chain length and M/G ratio.

Author Response

Thank you for the valuable comments and suggestions from the reviewer. We kindly request that you accept our responses and revisions.

Comment 1: Figure 1 presented unsolid results of response surface plots and this section needed to be repeated.

Response: Since this study was conducted in 2022, we no longer have any samples available for reanalysis. The data and figures presented were also included in my PhD thesis. We explored various ways to present the response surface plots and found the current format to be the most effective. We hope this presentation will be acceptable.

Comment 2: The recovery ratio, yield, and purity of alginate should be described.

Response: We have already provided the relevant data in the Results section and discussed these metrics in the text. For alginate purity, we have reported it as the alginate content (%) in the sample. We hope this is satisfactory. Unfortunately, further purity analysis cannot be performed due to the lack of available samples.

Comment 3: The alginate should be characterized structurally to confirm its composition, sugar chain length, and M/G ratio.

Response: Unfortunately, we no longer have the samples available to perform additional structural characterization. Therefore, we are unable to provide further details on the composition, sugar chain length, and M/G ratio.

Reviewer 3 Report

Comments and Suggestions for Authors

Dear Authors,

Here are some questions and issues regarding your paper content:

1. You have chosen as independent variables the sonication time and the US amplitude. Why didn't you also choose the frequency and temperature?

2. What about temperature rising during the sonication? was it monitored? how did you kept your samples at constant temperature (you do not mention any value of temperature in section 3.4.

3. The resulted math model looks good for alginate content. However the other two dependent variables (responses) didn't fit well. Could it be possible to choose first order polynomial equations instead second order polynomial equations or rather 2Fi type? Maybe separate modelling of each response would help. Simplifying the model by removing some terms may improve the values of R.

4. Figure 3 (FTIR spectra) contains some errors...instead of FTIR wavelength (cm-1), wavenumber (cm-1) should be placed. How were the spectra processed? The practice in FTIR spectroscopy is to display the values of frequency expressed as wavenumber in reverse order (from 1800 cm-1 to 500 cm-1).

5. Instead using HPLC acronym for the chromatography system used for the determination of the molecular weight distribution please use HPSEC or GPC. I agree that the only modification of the system is changing of the column (the mentioned used column is specially designed for size exclusion chromatography).Other particularity of GPC and HPSEC is refering of the data as plots of detector signal vs elution volume

6. I also strongly recommend showing calibration plots (in supplementary) and chromatograms revealing the molecular weight distribution of the separated alginates - the chromatograms would reveal any changes in the molar mass distribution. The PDI (polydispersity index) values may be computed as Mw/Mn to reveal the broadness of the distribution.

Kind regards,

Author Response

Many thanks to the reviewer for the valuable suggestions. We have made the changes in the manuscript and also highlighted them in red.

  1. You have chosen as independent variables the sonication time and the US amplitude. Why didn't you also choose the frequency and temperature?

Response:  Thank you for your question. In this study, we selected sonication time and ultrasound (US) amplitude as the independent variables for several reasons.

  • Firstly, these parameters are directly related to the energy input and intensity of the ultrasound treatment, which are crucial for optimizing the process. By focusing on these variables, we aim to achieve a more controlled and reproducible study, minimizing the potential for confounding factors.
  • While frequency and temperature are also important parameters, they were not chosen as independent variables in this study due to practical considerations and the specific objectives of our research. The equipment available to us operated at a fixed frequency, and the temperature was carefully controlled throughout the experiments. We also maintained the temperature of the extraction process using a circulation water bath at 30 °C.
  • Additionally, our industrial partners were particularly interested in exploring the effects of sonication time and amplitude, as they believed these factors would have the most significant impact on the process outcomes.
  • Lastly previous studies have demonstrated that sonication time and amplitude significantly influence the results of ultrasound treatments, providing a strong basis for our selection. By controlling these variables, we can maintain consistency and focus on the primary factors affecting the process.
  1. What about temperature rising during the sonication? was it monitored? how did you kept your samples at constant temperature (you do not mention any value of temperature in section 3.4.

Response:  Thank you for pointing it out. The temperature was controlled using a circulation water bath at 30 °C. This has now been mentioned in the text.  We did note before and after temperature and because of the temperature being controlled there was not much difference, and therefore was not mentioned in the manuscript.

  1. The resulted math model looks good for alginate content. However, the other two dependent variables (responses) didn't fit well. Could it be possible to choose first order polynomial equations instead second order polynomial equations or rather 2Fi type? Maybe separate modelling of each response would help. Simplifying the model by removing some terms may improve the values of R.

Response:  Thank you for your valuable feedback. We did consider using first-order polynomial equations and 2Fi (two-factor interaction) models for the other two dependent variables. However, after thorough analysis, we found that these approaches did not significantly improve the fit of the models. Additionally, we explored the possibility of separate modeling for each response and simplifying the models by removing some terms. Despite these efforts, the improvements in the values of R were not substantial enough to justify the changes.

Given these considerations, we decided to proceed with the current second-order polynomial equations. This approach provides a more comprehensive understanding of the interactions between variables and aligns with the objectives of our study. We believe that maintaining the current model structure will offer the most robust and insightful results for our research.

We appreciate your understanding and are open to any further suggestions you may have.

  1. Figure 3 (FTIR spectra) contains some errors...instead of FTIR wavelength (cm-1), wavenumber (cm-1) should be placed. How were the spectra processed? The practice in FTIR spectroscopy is to display the values of frequency expressed as wavenumber in reverse order (from 1800 cm-1to 500 cm-1).

Response: We agree that the FTIR wave number is a common way to mention it. The raw spectra were pretreated using baseline correction and mean centered. Wavelength has been replaced with wavenumber in lines 299, 458 and also in Figure 3.

  1. Instead using HPLC acronym for the chromatography system used for the determination of the molecular weight distribution please use HPSEC or GPC. I agree that the only modification of the system is changing of the column (the mentioned used column is specially designed for size exclusion chromatography). Other particularity of GPC and HPSEC is referring of the data as plots of detector signal vs elution volume

Response: Thank you for pointing it out. We have made the changes in the manuscript in the  lines 402, 405 and 414.

  1. I also strongly recommend showing calibration plots (in supplementary) and chromatograms revealing the molecular weight distribution of the separated alginates - the chromatograms would reveal any changes in the molar mass distribution. The PDI (polydispersity index) values may be computed as Mw/Mn to reveal the broadness of the distribution.

Response: Thank you for your recommendation. The work in question was conducted during my PhD in 2022, and unfortunately, I no longer have access to the data or equipment as I am now based in a different country. However, I appreciate the importance of including calibration plots and chromatograms to reveal the molecular weight distribution and PDI values. I will ensure to incorporate these elements in my upcoming work to provide a more comprehensive analysis.

Many thanks again.

Round 2

Reviewer 1 Report

Comments and Suggestions for Authors

All questions were answered well.

Author Response

Thank you so much. 

Reviewer 2 Report

Comments and Suggestions for Authors

rejected.

Author Response

Thank you for time to review our paper. 

Reviewer 3 Report

Comments and Suggestions for Authors

Dear Authors,

I agree with most of your modifications. There is still a problem regarding figure 3  X axis which shows FTIR wavenumber in cm-1. The correct form is simply Wavenumber (cm-1). In general the FTIR spectra is displayed from higher wavenumber to lower ... an example is   https://photometrics.net/fourier-transform-infrared-ftir-spectroscopy/

and also here...Under this condition your x axis will start at about 1800 cm-1 and end at about 400-500. cm-1

Author Response

Thank you for pointing out that the FTIR data is presented in decreasing order of wavenumber. It is also a common practice to present it in increasing order, there are numerous publications that present data in increasing wavenumber formats. 

doi: 10.5812/jjm.28320

https://doi.org/10.3390/ijms23105410

The author who conducted the FTIR work is no longer available to make this correction.  We appreciate your understanding and will ensure that such an oversight is not repeated in future publications. Hope it can be ignored this time. 

Round 3

Reviewer 2 Report

Comments and Suggestions for Authors

Accept.